# Ex Vivo Antiplatelet and Thrombolytic Activity of Bioactive Fractions from the New-Fangled Stem Buds of *Ficus religiosa* L. with Simultaneous GC-MS Examination

**DOI:** 10.3390/molecules28093918

**Published:** 2023-05-05

**Authors:** Sunil Kumar, Muhammad Arif, Mehnaz Kamal, Talha Jawaid, Mohammed Moizuddin Khan, Beenish Mukhtar, Abdullah Khan, Saif Ahmed, Saud M. AlSanad, Osama A. Al-Khamees

**Affiliations:** 1Department of Pharmacognosy, Faculty of Pharmacy, Integral University, Kursi-Road, Lucknow 226026, Uttar Pradesh, India; sunilkumar39781@gmail.com; 2Department of Pharmaceutical Chemistry, College of Pharmacy, Prince Sattam bin Abdulaziz University, Al- Kharj 11942, Saudi Arabia; mailtomehnaz@gmail.com; 3Department of Pharmacology, College of Medicine, Imam Mohammad Ibn Saud Islamic University (IMSIU), Riyadh 13317, Saudi Arabia; tjkhan@imamu.edu.sa (T.J.); alsanads@gmail.com (S.M.A.); oalkhamees@hotmail.com (O.A.A.-K.); 4Department of Basic Medical Science, College of Medicine, Dar Al Uloom University, Riyadh 13314, Saudi Arabia; moizuddin@dau.edu.sa (M.M.K.); bmukhtar@dau.edu.sa (B.M.); 5Department of Physiology, Santosh Deemed to be University, Ghaziabad 201009, Uttar Pradesh, India; 6Faculty of Pharmacy, Quest International University, Ipoh 30250, Malaysia; abdullah.khan@qiu.edu.my; 7Department of Physiology, College of Medicine, Imam Mohammad Ibn Saud Islamic University (IMSIU), Riyadh 13317, Saudi Arabia; asahmed@imamu.edu.sa

**Keywords:** ex vivo study, antiplatelet and thrombolytic activity, new-fangled stem buds, *Ficus religiosa*, toxicological effects, GC-MS examination

## Abstract

Different parts of *Ficus religiosa* are the common components of various traditional formulations for the treatment of several blood disorders. The new-fangled stem buds’ powder was extracted with 80% ethanol and successively fractionated by chloroform and methanol. Chloroform and methanol fractions of *Ficus religiosa* (CFFR and MFFR) were tested for antiplatelet, antithrombotic, thrombolytic, and antioxidant activity in ex vivo mode. The MFFR was particularly investigated for GC-MS and toxicity. The antiplatelet activity of the CFFR, MFFR, and standard drug aspirin at 50 μg/mL was 54.32%, 86.61%, and 87.57%, and a significant delay in clot formation was noted. CFFR at different concentrations did not show a significant effect on the delay of clot formation, antiplatelet, and free radical scavenging activity. The most possible marker compounds for antiplatelet and antioxidant activity identified by GC-MS in the MFFR are salicylate derivatives aromatic compounds such as benzeneacetaldehyde (7), phenylmalonic acid (13), and Salicylic acid (14), as well as Benzamides derivatives such as carbobenzyloxy-dl-norvaline (17), 3-acetoxy-2(1H)-pyridone (16), and 3-benzylhexahydropyrrolo [1,2-a] pyrazine-1,4-dione (35). A toxicity study of MFFR did not show any physical indications of toxicity and mortality up to 1500 mg/kg body weight and nontoxic up to 1000 mg/kg, which is promising for the treatment of atherothrombotic diseases.

## 1. Introduction

Thrombosis is a common pathology of basic ischemic heart ailment, stroke, acute coronary syndrome, acute myocardial infarction, angina, primary and secondary cardiovascular disease, and venous thromboembolism. The Global Burden of Disease Study 2020 (GBD 2020) recognized that ischemic heart ailment affects around 1.72% of the world’s population [1]. Platelets are important to the beginning of thrombosis, and drugs that disturb the platelet function used in the management of thrombotic disorders are known as antiplatelet drugs [2,3]. There are numerous antiplatelet drugs existing for use in medical practice and several are under study. Current drugs such as aspirin, ticagrelor, clopidogrel, apixaban, and warfarin are valuable; however, they have unwanted toxicological effects, which include extended bleeding time, purpura, thrombocytopenia, and gastrointestinal tracts ulcer. Major side effects of chemotherapy detected in clinical use are diminishing as is clampdown of the immune system [4]. With attention to the usefulness to-care ratio in drug design and assortment, there is a necessity to search for active antiplatelet agents with negligible or no side effects. In such clinical conditions, herbal drugs having antioxidant effects may be of beneficial therapeutical use in the treatment of patients [5]. Herbal drugs are a better supernumerary medicine in contrast to synthetic medicine due to their affordability and effectiveness. Herbal drug investigator phenolics compounds, triterpenoids, phytosterols, and vitamins show an extensive range of biological actions and these are primarily a result of their antioxidant properties [6].

*Ficus religiosa*, generally named “Peepal” and bodhi tree, is a huge, widely ramous tree accompanied by mythical, religious, and medicinal significance in Indian culture [7]. Leaves, fruits, and bark of the plant contain lanosterol, β–sitosterol, bergaptol, bergapten, steroids, stigmasterol, lupen-3-one, vitamin K1, tannin, wax, saponin, leucoanthocyanidin, leucoanthrocyanin, and leucopelargonidin [8]. Traditionally, it has been exploited to treat a diversity of maladies, such as hypertension, diabetes, dysentery, epilepsy, gastrointestinal issues, inflammatory problems, infections, blood disorders, and sexual dysfunction [9]. The bioactive components, such as phenolics, flavonoids, steroids, triterpenoids, and their glycosides, accumulate in the budding parts of the flower and the stem buds to attract insects and other agents for pollination [10]. The bioactive phenolic compounds, polyphenols, and flavonoids are present in large quantities in these developing parts (Figure 1) of the plants, which are potent natural antioxidants and provide anti-inflammatory, antiplatelet, antithrombotic, and thrombolytic activity. The stem bud is excellent for the management of thrombotic disorders [11]. This research intends to discover and analyze the phytoconstituents of the new-fangled stem bud *F. religiosa* by GC-MS and evaluate it for antiplatelet, antithrombotic, thrombolytic, and toxicological activity.

## 2. Results

Preliminary phytoconstituents testing of the CFFR in the stem bud extract showed the presence of steroids and triterpenoids whereas the MFFR confirms the presence of alkaloids, reducing sugar, flavonoids, and phenolic compounds. So, both the CFFR and the MFFR of the stem bud extracts were designated for the study.

### 2.1. Antiplatelet Activity Testing

The antiplatelet action of *F. religiosa* was based on absorbance taken by a UV spectrophotometer where collagen was used as a platelet aggregation persuader. The +ve control (aspirin) and −ve control (normal saline) highlighted the suitability of the procedures in the study. The comparative antiplatelet activities of CFFR, MFFR, and aspirin at various concentrations are tabulated in Table 1. At different concentrations, the MFFR showed excellent antiplatelet activity whereas the CFFR did not show any activity. The minimum concentration exhibiting antiplatelet activity in the MFFR was 10 μg/mL. The highest used concentration of the MFFR was 50 μg/mL, which shows dose-dependent activity. The antiplatelet activity of the MFFR at 40 μg/mL is equivalent to 16 μg/mL of aspirin.

### 2.2. Thrombolytic and Antithrombotic Activity

Comparative thrombolytic and antithrombotic activities of CFFR, MFFR, and SK are shown in Table 2. Clot lysis was not observed in the negative control (tube-1). In tube-10 containing SK, clot lysis was seen within 40–50 min., and in the MFFR, significantly dose-dependent clot lysis was observed whereas the CFFR did not show a significant effect on clot dissolution. Maximum clot lysis occurred in tube-9, which contained a 40 µg/mL concentration of MFFR. In the observation of antithrombotic activity, the clot was formed in normal time after the addition of NS to the control (tube-1). Whereas in tube-10 containing SK, the clot was not found, and in the case of various concentrations of MFFR, a significant adjournment in clot formation was observed. The maximum delay in clot development was noted at 40 µg/mL in tube-9 whereas CFFR at different concentrations did not show an effect on the delay of clot formation.

Antithrombotic and thrombolytic activity is presented for CFFR, MFFR, and SK and monitoring was carried out for up to 60- and 90-min. Clot formation of SK (positive control) in the case of antithrombotic activity did not occur in 60 min. Clot dissolution of SK (positive control) occurred within 60 min. and in CFFR it had not occurred in 90 min.

### 2.3. Effect of Stem Bud Extract on the Peroxidation of Linoleic Acid

Comparative peroxidation of linoleic acid by CFFR, MFFR, and ascorbic acid are shown in Figure 2. Results observed from the ferric thiocyanate test reveal the MFFR fraction of *F. religiosa* stem-bud extract brings the antioxidant activity for the chain-disrupting of lipid peroxidation. Both ascorbic acid and MFFR showed significant dose-dependent free radical scavenging activity whereas CFFR exhibited nonsignificant activity. Both ascorbic acid and MFFR at a dose of 300 µg/mL showed maximum 12.43%, 78.1%, and 96.4% inhibition of free radical scavenging activity. The IC_50_ values of the MFFR and ascorbic acid were 147.28 ± 0.57 and 98.14 ± 0.66 µg/mL.

### 2.4. Toxicity Studies

A toxicity study of the methanol fraction (MFFR) did not show any physical indications of toxicity and mortality up to 1500 mg/kg body weight. Administration of MFFR at 1500 mg/kg b.w. produced a significant (*p* < 0.05) decrease in red blood cells (RBCs), white blood cells (WBCs), hemoglobin (Hb), body weight, and liver weight whereas there were increases in the spleen weight. There was no effect on heart and kidney weight. The weight of the liver, kidney, spleen, heart, and the content of RBCs, leukocytes, and Hb were normal at 500 mg/kg b.w. and had nonsignificant changes at 1000 mg/kg b.w. as compared to the control group (Table 3 and Table 4).

### 2.5. GC-MS Analysis of Methanol Fraction of Stem Bud Extract

The objective of the GC-MS analysis of the MFFR was to investigate the pharmacologically dynamic components present in the methanol fraction of the stem bud extract of *F. religiosa*. The result shows the existence of different polar and non-polar phytocomponents. In the MFFR, a total of 37 classes of phytoconstituents were identified, which are listed in Table 5 with retention time, percentage area, compound name, and nature of compounds. The molecular structure of the chemical constituents, molecular weight, and molecular formula are in Figure 3. The prevailing compounds identified in the MFFR were a major polar component and a few non polar components, such as 3-Methylcyclopentyl acetate (0.52%), 2-Hydroxypropanoic acid (5.01%), 2-Methyl-N-[(E)-2-methylbutylidene]-1-propanamine (0.38%), 3-Methylenedihydro-2,5-furandione (0.66%), 1,2,3,4-Butanetetrol (1.77%), 1-Butanamine, 2-methyl-N-(2-methylbutylidene) (1.12%), Benzeneacetaldehyde (2.41%), Succinic acid, monomethyl ester (2.54%), 3,5-Dihydroxy-6-methyl-2,3-dihydro-4H-pyran-4-one (1.76%), Dehydromevalonic lactone (0.68%), Benzoic acid (1.78%), 2,5-Dimethyl-2-hexanol (1.48%), Phenylmalonic acid (1.89%), Salicylic acid (3.74%), 2-Acetamido-2-deoxymannosonic acid (4.16%), 3-Acetoxy-2(1H)-pyridone (1.99%), Carbobenzyloxy-dl-norvaline (0.78%), 4-(1,3,3-Trimethyl-bicyclo[4]hept-2-yl)-but-3-en-2-one (0.70%), 3,4,5,6,7,8-Hexahydro-1(2H)-naphthalenone (1.29%), 3H-3,10A-methano-1,2-benzodioxocin-3-ol (0.40%), 1,3,4,5-tetrahydroxy-cyclohexanecarboxylic acid (4.65%), Ingol-12-acetat (0.24%), (2E)-3,7,11,15-Tetramethyl-2-hexadecene (1.19%), 2,2-Dimethyl-1-oxaspiro[2.5]octan-4-one (2.06%), Pluchidiol (11.86%), Cyclo-L-prolyl-L-valine 3.35%), 2-Methoxy-4,4-dimethyl-2-cyclohexen-1-one (1.67%), 3,3-Dimethylglutaric acid (2.14%), l-Leucine, N-cyclopropylcarbonyl-, dodecyl ester (1.16%), 3-Isobutylhexahydropyrrolo[1,2-a]pyrazin (1.59%), n-Hexadecanoic acid (4.08%), Xanthaumin (5.51%), 3-Methylene-bicyclo[3.2.1]oct-6-en-8-ol (0.63%), (+/−)-Marmesin (0.97%), 3-Benzylhexahydropyrrolo[1,2-a]pyrazine-1,4-dione (0.71%), 4,6-Cholestadien-3.beta.-ol (1.09%), Lup-20(29)-en-3.beta.-ol, acetate (1.36%). The GC-MS report can be used for authorizations of the different biochemical markers in the plant’s extract.

## 3. Discussion

The antiplatelet activity results show that the methanol fraction of the stem bud extract (MFFR) prevents collagen-induced platelet aggregation in a dose-dependent manner in ex vivo mode whereas CFFR did not show a significant effect. The working collagen concentration of 0.2 mL (2 µg/mL) used in the experiment was the optimized quantity as stated in the published work [12]. The antiplatelet activity of the MFFR at 40 μg/mL is equivalent to 16 μg/mL of aspirin. The antiplatelet (cardioprotective effect) consequence of salicylate is arbitrated by the reduction of the production of thromboxane and the irreversible embarrassment of platelet cyclo-oxygenase (COX-1). Benzeneacetaldehyde, salicylic acid, phenylmalonic acid and 1,3,4,5-tetrahydroxy-cyclohexanecarboxylic acid present in MFFR might act as irreversible COX-1 inhibitors such as aspirin because they have a similar structure to the standard drug 2-Acetoxybenzoic acid (aspirin) [13]. Diundecyl phthalate and l-Leucine, and N-cyclopropylcarbonyl-dodecyl ester is structurally similar to the Thromboxane inhibitors that are a potential antiplatelet category of drugs [14].

Similarly, 3-Acetoxy-2 (1H)-pyridone, Carbobenzyloxy-dl-norvaline, 3-Isobutylhexahydropyrrolo[1,2-a] pyrazin, and 3-Benzylhexahydropyrrolo[1,2-a]pyrazine-1,4-dione are chemically similar to the benzamide derivatives shown in Figure 4. The presence of new benzamide derivatives, benzimidazole, and 2-Acetoxybenzoic acid derivatives in the structures of the component might show a potential antiplatelet effect in the ex vivo model, which significantly shows similar activity to the reference drug, aspirin [15].

Nucleic acids, proteins, and bio-membranes, of a cell can be damaged by the oxidation of free radicals, which increases lipid peroxidation [16]. It initiates the impairment of phospholipids and lipoproteins in the cell membrane by proliferating a chain reaction cycle started by a reactive oxygen species [17]. The unsaturated cyclic compounds with -OH groups play a key role in the free radical scavenging activity. The polar compounds such as 3-Methylcyclopentyl acetate, Benzeneacetaldehyde, Succinic acid, 2,5-Dimethyl-2-hexanol, Phenylmalonic acid, Salicylic acid, 2-Acetamido-2-deoxymannosonic acid, Ingol-12-acetate, Pluchidiol, and Xanthaumin might be a very good antioxidant component. This possibility is studied in the test of ascorbate-dependent lipid peroxidation (LPO). The strong antioxidant activity of these components present in the MFFR of the plant will help counter oxidative stress-related disorders [18]. The use of antioxidants from natural resources captures the ROS to enrich oxidative stress-induced ailments.

Numerous types of research have led to the investigation of the antithrombotic and thrombolytic activity of traditional drug supplements. Traditionally consuming herbal-based supplements leads to preventing atherothrombotic ailments. This study on the antithrombotic and thrombolytic activity of MFFR exhibited antithrombotic activity and delayed clot formation in a dose-dependent manner. Chloroform fraction CFFR had a non-significant effect. MFFR was freely solubilized in normal saline and exhibited significant antithrombotic and thrombolytic activity whereas CFFR was less dissolved in normal saline and expression less activity.

The variations in Hb, RBC, WBC, and organ and body weight are generally observed in patients during chemotherapy [19]. The toxicological results of MFFR demonstrated that treatment with a dose of 1500 mg/kg b.w. produced a significant (*p* < 0.05) decrease in the total RBCs, leukocytes, Hb, body weight, and liver weight. The weight of the liver, kidney, spleen and heart as well as the content of RBCs, leukocytes, and Hb were normal at 500 mg/kg b.w. and were nonsignificant at 1000 mg/kg b.w. as compared to the control group. It benefits through a scientific confirmation of native use and provides a chance to use recent data to produce traditional herbal drugs that are unique, harmless, and effective.

We suggest that the free radical scavenging effects of the MFFR might prevent collagen-prompted platelet aggregation without any harm and may help alleviate or reduce the development of numerous oxidative stress-related ailments. It could be a capable plan in addressing many cardiovascular ailments such as thrombosis, atherosclerosis, and myocardial infarction.

## 4. Materials and Methods

### 4.1. Plant Material

The new-fangled stem buds were plucked from the tree *F. religiosa* in February 2021 located on the side of Kursi Road, Lucknow, UP, India, and authenticated from NBRI, Lucknow. Receipt no. LWG-109223 was reserved for Appendix A.

### 4.2. Chemicals and Drugs

Analytical grade chemicals were used in the project, such as solvents for extraction and fractionation gained from Merck (Mumbai, India). Streptokinase (SK) was available as a lyophilized SK vial and aspirin was obtained as a gift sample from Cadila Pharma.

### 4.3. Instruments

UV Spectrophotometer Shimadzu-U Singapore, GC-MS apparatus comprises Shimadzu QP-2010 Ultra through capillary standard non-polar 60 M TRX-5 MS column.

### 4.4. Extraction and Fractionation

The new-fangled stem buds of *F. religiosa* were shade dried under a blower and pulverized. The powdered plant material (250 g) was macerated with 600 mL of 80% *v*/*v* ethanol at room temperature for 72 h. The extract obtained by maceration was filtered and the filtrate was evaporated under reduced pressure. The extract yield was 9.7 % *w*/*w*. The dried ethanol extract was successively fractionated with the column by eluting the chloroform and methanol. The percentage yields of chloroform and methanol fractions of *Ficus religiosa* were 1.30 and 7.65% *w*/*w* respectively [20]. Both chloroform and methanol fractions were also subjected to preliminary phytochemical examination for the existence of various phytoconstituents [21].

### 4.5. Blood Sample

Laboratory-bred Sprague-Dawley male and female (150–200 g, 10- to 12-week-old) rats were selected for the antiplatelet and thrombolytic activity. The experimental animals were procured from the Central Drug Research Institute, Lucknow, India and reserved under standard laboratory environments (temperature 23 ± 2 °C, R.H. 50 ± 15%, and 12 h dark/light) in the animal house of the Faculty of Pharmacy, Integral University, Lucknow. Suitable food pellets and water were given ad libitum for seven days. Seven days later, acclimated rat blood was collected through the retro-orbital sinus under light anesthesia. The tentative procedure was agreed upon by the Institutional Ethical Committee (Endorsement no. IU/IAEC/21/11) following the guidelines of the Committee for the Purpose of Control and Supervision of Experiments on Animals (CPCSEA).

### 4.6. Antiplatelet Activity Testing

In this method, collagen (agonist) was added to platelet-rich plasma (PRP), which was accumulated, condensed, and exposed to less light, so the transmission was amplified, which was sensed by the UV spectrophotometer. The PRP was produced by the centrifugation of rat blood (comprising 0.9% Na-citrate as anti-coagulant) at 1700 rpm for 10 min. The strength of the PRP has attuned to OD 1.0 at λ*max* 425 nm by the addition of normal saline. The chloroform and methanol fractions of the stem bud extract (CFFR and MFFR) were dissolved in normal saline in increasing concentrations of 5, 10, 15, 20, 25, 30, 35, 40, and 50 μg/mL, and standard drug aspirin in concentrations of 2, 4, 6, 8, 10, 12, 14, 16, and 18 μg/mL (Tube no. 2–10). Tube no. 1 held normal saline as a negative control. The PRP (1.5 mL) was added to all 10 tubes and the final volume of each tube was brought up to 2.5 mL by the addition of normal saline. All the tube mixtures were incubated at 37 ± 2 °C for 10 min and 0.2 mL (2 μg /mL) of collagen was added to all tubes. The aggregation of platelets was tempted under constant stirring at 700 rpm for 4 min. and the aggregation was observed through a UV spectrophotometer at λ 560 nm [22,23].

### 4.7. Thrombolytic and Antithrombotic Activity

Experimentations for clot dissolutions and antithrombotic action of fractions CFFR and MFFR from ethanol extract of *F. religiosa* new-fangled stem buds, and standard drug Streptokinase (SK) was carried out. Different concentrations (5, 10, 15, 20, 25, 30, 35, and 40 μg/mL) of test samples (CFFR and MFFR) were incorporated to clean microcentrifuge tubes holding 0.5 mL of blood in all tube no. 2–8. Tube no. 1 contained normal saline solution as a negative control. Streptokinase (SK) of 1,500,000 I.U. appropriately sorted in sterilized normal saline was used as a positive control. The contents of all tubes were adjusted to 1 mL and time was renowned before each incorporation of the tested sample into the 0.5 mL of blood. The tested samples (CFFR, MFFR, SK, and NS) were added into the marked tubes before clot formation after taking blood from the tubes for antithrombotic action and for thrombolytic action. The tested samples were incorporated just after clot formation. All reactions were sustained at 36 ± 2 °C and all the samples of each concentration were tested 3 times [24,25].

### 4.8. Effect of Stem Bud Extract on Linoleic Acid Peroxidation

For testing the peroxyl radical scavenging activity, different concentrations of CFFR, MFFR, and ascorbic acid (50, 100, 150, 200, 250, and 300 µg/mL) were mixed in 0.5 mL of double distilled water and mixed with 2.5 mL of 0.02 M linoleic acid in 2 mL of 0.04 M phosphate buffer (pH 7.0) in a test tube and gestated in the dark at 36 ± 2 °C. During the incubation period, the quantity of peroxide generated was determined by an analysis of the absorbance of color at 540 nm by mixing 0.1 mL 35% ammonium thiocyanate and 0.1 mL 20 mM ferrous chloride in 3.5% HCl to the reaction combination. A control was correspondingly prepared to replace the test sample with water [16].

### 4.9. Toxicity Studies

The toxicity study was accomplished by following OECD rule No. 452 [26]. Sprague-Dawley rats of either sex were separated into four groups with six rats in each group. The methanol fraction of the stem bud extract (MFFR) was given orally at a single dose level of 500, 1000, and 1500 mg/kg b.w. All the animals were watched intermittently for symptoms of toxicity and mortality within 24 h and then daily for 15 days. On the 15th day, the rats were anesthetized by light ether and blood was withdrawn by cardiac puncture. The vital body parts, such as the liver, kidney, spleen, and heart, were removed, cleaned, and blotted and then relative organ weight was determined. The content of red blood cells (RBC), white blood cells (WBC), and hemoglobin (Hb) were determined by Hayem’s, Turke’s, and Sahli’s methods [27,28].

### 4.10. GC-MS Analysis of Methanol Fraction of Stem Bud Extract

The methanol fraction of the stem bud extract (MFFR) was subjected to gas chromatography-mass spectroscopy (GC-MS). Helium was utilized as the carrier gas and the flow amount of carrier gas was set at 1.21 mL per min. The temperature of the instrument system was elevated from 100 to 260 °C at an increment of 10°C per min. The injecting volume was adjusted to 2 μL. An electron ionization energy scheme with an ionization energy of 70 eV was utilized for GC-MS detection. The sample MFFR was mixed in methanol and ran fully at a range of 10–870 *m*/*z* for 30 min. The outcomes were compared utilizing the Wiley spectral library search program and mass spectra were detected in 0–30 min. The compound name, molecular weight, formula, and structure were determined. The relative percentage of all compounds was intended by comparison of its middling peak area to the total areas. Each compound’s proof of identity was supported by a qualitative chemical test, comparing the retention time and spectra with reliable samples (Sigma-Aldrich, St. Louis, MO, USA) and software adapted to handle mass spectra in the NIST Mass Spectral Library Ver. 2.0 d [29].

### 4.11. Statistical Analysis

The experimental data are presented as the mean ± standard deviation. One-way analysis of variance trailed by the Dunnett test was accomplished using the Graph Pad Prism 2.01 (Graph Pad Software, Inc., San Diego, CA, USA), and a value of *p* < 0.05 and *p* < 0.01 was measured as statistically significant.

## 5. Conclusions

From the results of the study, it can be concluded that the new-fangled stem buds of *Ficus religiosa* showed antiplatelet, antithrombotic, thrombolytic, and antioxidant activity and that the MFFR has the ability to avert cardiovascular disease. The MFFR may be incorporated as an antiplatelet, antithrombotic, and thrombolytic agent for the improvement of patients suffering from atherothrombotic diseases. The MFFR comprises most of the polar phytoconstituents of the selected plant extract, which was qualitatively authenticated by the GC-MS. Most of the compounds identified by the GC-MS in the MFFR (Benzeneacetaldehyde, Salicylic acid, Phenylmalonic acid and 1,3,4,5-tetrahydroxy-cyclohexanecarboxylic acid, Diundecyl phthalate and l-Leucine, and N-cyclopropylcarbonyl-dodecyl ester) might be responsible for potential antiplatelet activity through selective COX-1 and thromboxane inhibitor pathways. The MFFR is safe up to the dose of 1000 mg/kg b.w. and may be utilized for the treatment of patients suffering from atherothrombotic diseases. The investigation discovered the varied medicinal compounds that will be beneficial to the pharmaceutical segment utilizing various phytoconstituents in the treatment of atherothrombotic diseases.

## Figures and Tables

**Figure 1 molecules-28-03918-f001:**
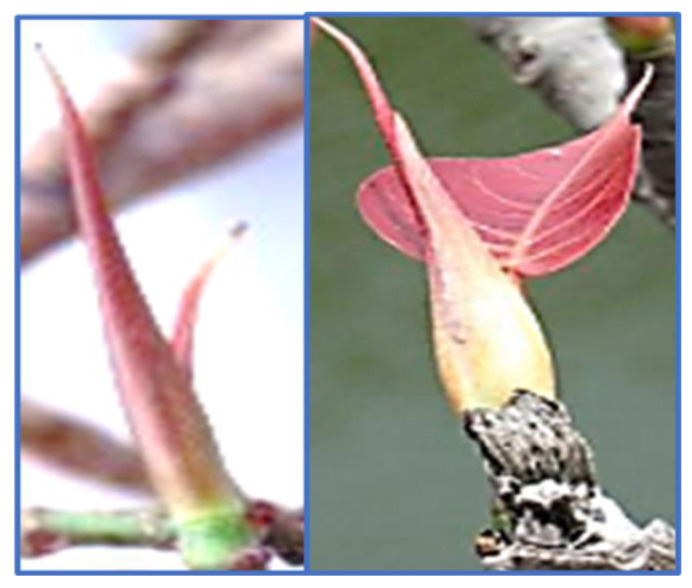
New-fangled stem buds of *Ficus religiosa*.

**Figure 2 molecules-28-03918-f002:**
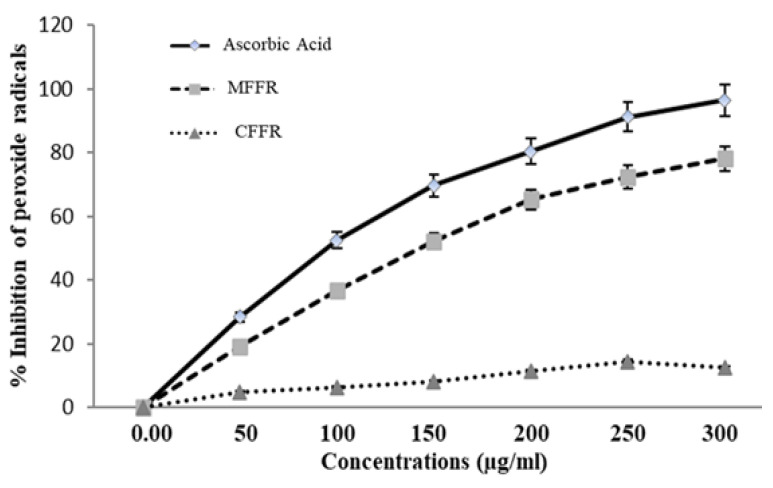
Dose-dependent inhibition of lipid peroxidation by the chloroform and methanol fraction of new-fangled stem buds of *Ficus religiosa*, and standard ascorbic acid at concentrations (50, 100, 150, 200, 250, and 300 µg/mL). Each value represents mean ± S.D. (*n* = 3). IC_50_ = 147.28 ± 0.57 for MFFR and 100.14 ± 0.66 µg/mL of the standard ascorbic acid.

**Figure 3 molecules-28-03918-f003:**
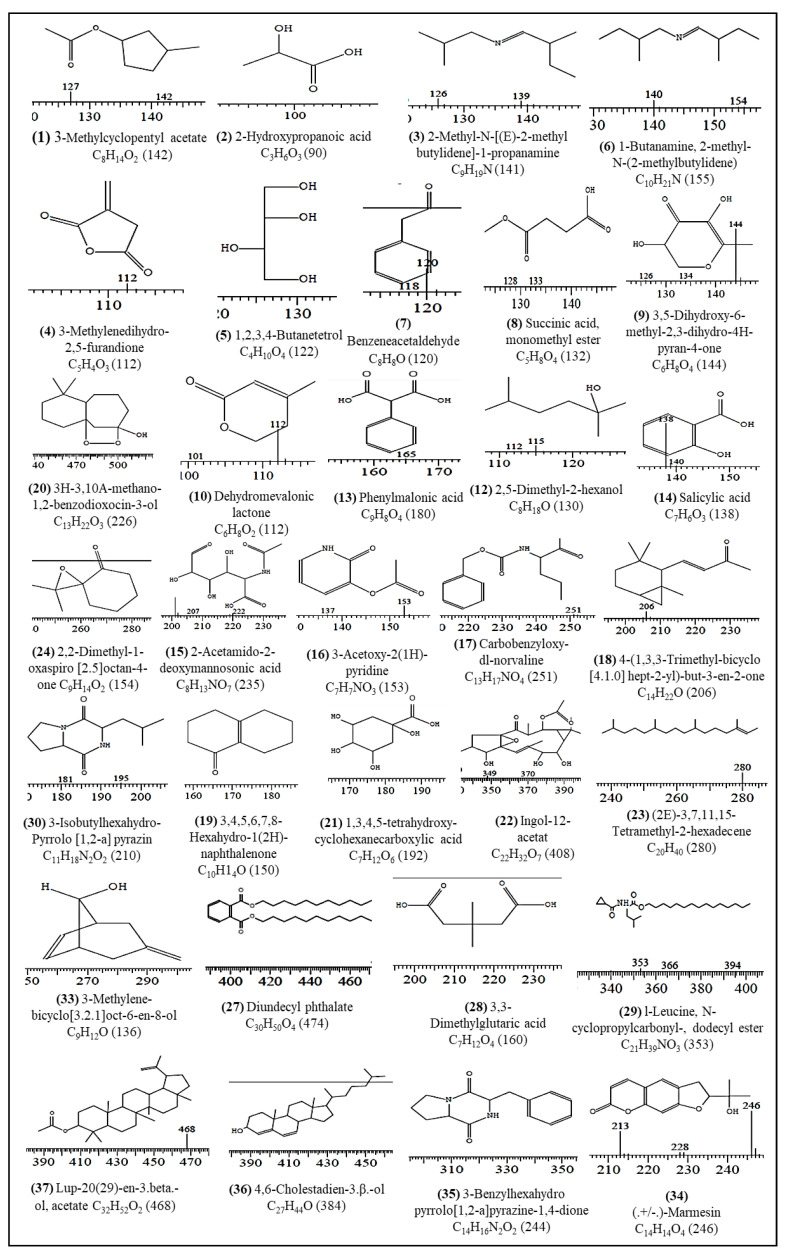
Chemical compounds hit upon the GC-MS chromatogram of the MFFR extract.

**Figure 4 molecules-28-03918-f004:**
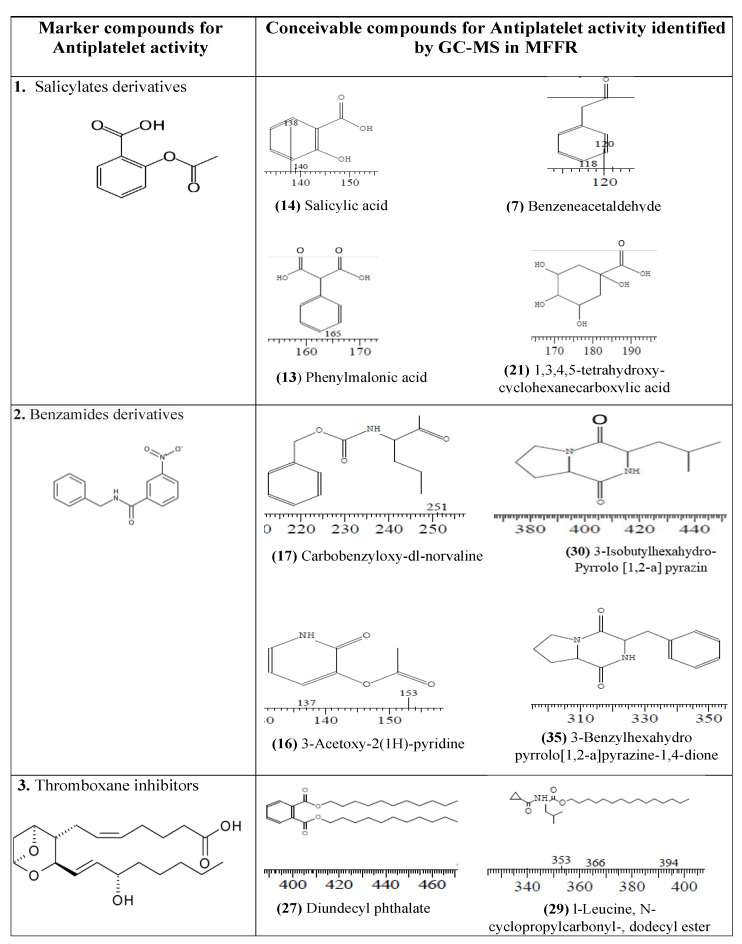
Conceivable marker compounds for antiplatelet activity identified by GC-MS in MFFR.

**Table 1 molecules-28-03918-t001:** Antiplatelet activity of chloroform and methanol fractions of the new-fangled stem bud extract of *Ficus religiosa*.

Tube No.	Conc. of Aspirin (µg/mL)	Conc. of Test Sample (µg/mL)	Absorbance		
			CFFR	Aspirin	MFFR
1.	0.0	0.0	0.119 ± 0.021	0.119 ± 0.021	0.119 ± 0.021
2.	2	5	0.123 ± 0.035 ^ns^	0.194 ± 0.054 *	0.164 ± 0.058 ^ns^
3.	4	10	0.134 ± 0.027 ^ns^	0.236 ± 0.047 *	0.218 ± 0.026 *
4.	6	15	0.146 ± 0.047 ^ns^	0.394 ± 0.074 *	0.349 ± 0.048 *
5.	8	20	0.159 ± 0.058 ^ns^	0.517 ± 0.038 *	0.478 ± 0.042 *
6.	10	25	0.166 ± 0.031 ^ns^	0.594 ± 0.038 *	0.539 ± 0.053 *
7.	12	30	0.184 ± 0.062 ^ns^	0.686 ± 0.075 *	0.611 ± 0.062 *
8.	14	35	0.199 ± 0.029 ^ns^	0.791 ± 0.069 **	0.698 ± 0.055 *
9.	16	40	0.217 ± 0.044 *	0.887 ± 0.048 **	0.744 ± 0.064 **
10.	18	50	0.259 ± 0.050 *	0.958 ± 0.056 **	0.889 ± 0.059 **

Data were articulated as mean standard deviation (*n* = 3). One-way analysis of variance was accomplished followed by the Dunnett t-test. * *p* < 0.05 and ** *p* < 0.01 related to resting platelets (negative control) and “ns” (No significant value).

**Table 2 molecules-28-03918-t002:** Antithrombotic and thrombolytic activity of chloroform and methanol fractions of the new-fangled stem bud extract of *Ficus religiosa*.

Sample No.	Antithrombotic Activity			Thrombolytic Activity		
	Conc. of Tested Sample (µg/mL)	Clot Formation Delayed Time (min.)		Conc. of Tested Sample (µg/mL)	Clot Dissolution Time (min.)	
		CFFR	MFFR		CFFR	MFFR
1	0.0 (Blank)	4.20	4.20	0.0 (Blank)	NO	NO
2	5	4.50	10.40	5	NO	NO
3	10	5.10	13.20	10	NO	NO
4	15	5.40	15.40	15	NO	NO
5	20	6.20	18.30	20	NO	82 ^#^
6	25	6.50	22.20	25	NO	75 ^#^
7	30	7.80	26.40	30	NO	67 ^#^
8	35	8.10	29.30	35	NO	58 ^#^
9	40	8.30	41.20	40	NO	52 ^#^
10	SK	NO *	NO *	SK	46 ^#^	46 ^#^

***** More than 60 min, **^#^** More than 90 min.

**Table 3 molecules-28-03918-t003:** Effect of methanol fraction of stem bud extract (MFFR) on changes in body weight, Hb, RBCs and WBCs count.

Treatment Group	Changes in Body Weight	Hb (g/dL)	RBC Count 10^6^/µL	WBC Count Cells/mm^3^
Control vehicle 5 mL/kg	162.79 ± 0.86	16.40 ± 0.071	5.41 ± 0.096	9162 ± 2156
MFFR 500 mg/kg	161.27 ± 0.83	16.35 ± 0.086	5.42 ± 0.081	9126 ± 2115
MFFR 1000 mg/kg	154.72 ± 0.84	13.18 ± 0.087	4.71 ± 0.197	8814 ± 2154
MFFR 1500 mg/kg	148.76 ± 0.77 *	11.18 ± 0.126 *	4.35 ± 0.089 *	7872 ± 2168 *

Values are expressed as the mean ± S.E.M. of six rats in each group. One-way analysis of variance (ANOVA) and Dunnett’s were performed. * *p* < 0.05 compare to the respective control group.

**Table 4 molecules-28-03918-t004:** Effect of methanol fraction of stem bud extract (MFFR) on changes of the relative organ weight of rats.

Treatment Group	Liver (g/1000)	Spleen (g/1000)	Kidney (g/1000)	Heart (g/1000)
Control vehicle 5 mL/kg	12.14 ± 0.05	1.28 ± 0.09	4.14 ± 0.11	7.15 ± 0.09
MFFR 500 mg/kg	12.17 ± 0.08	1.37 ± 0.16	4.16 ± 0.15	7.11 ± 0.14
MFFR 1000 mg/kg	10.12 ± 0.09	1.46 ± 0.12	4.21 ± 0.12	6.78 ± 0.09
MFFR 1500 mg/kg	08.16 ± 0.07 *	1.54 ± 0.10 *	4.15 ± 0.14	6.46 ± 0.11 *

Values are expressed as the mean ± S.E.M. of six rats in each group. One-way analysis of variance (ANOVA) and Dunnett’s were performed. * *p* < 0.05 compare to the respective control group.

**Table 5 molecules-28-03918-t005:** Compounds identified in MFFR of the new-fangled stem bud extract of *Ficus religiosa* through GS-MS.

S. No.	R. Time	Area%	Compound Name	Nature of Compounds
1	4.325	0.52	3-Methylcyclopentyl acetate	Ester
2	4.450	5.01	2-Hydroxypropanoic acid	Fatty acid
3	4.841	0.38	2-Methyl-N-[(E)-2-methylbutylidene]-1-propanamine	Amine
4	5.019	0.66	3-Methylenedihydro-2,5-furandione	Heterocyclic compounds
5	6.305	1.77	1,2,3,4-Butanetetrol	Erythritol
6	6.486	1.12	1-Butanamine, 2-methyl-N-(2-methylbutylidene)	Amine
7	6.660	2.41	Benzeneacetaldehyde	Aromatic compound
8	7.785	2.54	Succinic acid, monomethyl ester	Ester
9	8.347	1.76	3,5-Dihydroxy-6-methyl-2,3-dihydro-4H-pyran-4-one	Sugar
10	8.549	0.68	Dehydromevalonic lactone	Lactone
11	8.799	1.78	Benzoic acid	Aromatic acid
12	9.524	1.48	2,5-Dimethyl-2-hexanol	Alcohol
13	9.986	1.89	Phenylmalonic acid	Aromatic acid
14	10.837	3.74	Salicylic acid	Aromatic acid
15	11.214	4.16	2-Acetamido-2-deoxymannosonic acid	Amide compound
16	11.877	1.99	3-Acetoxy-2(1H)-pyridone	Aromatic compound
17	12.358	0.78	Carbobenzyloxy-dl-norvaline	Aromatic compound
18	12.611	0.70	4-(1,3,3-Trimethyl-bicyclo[4]hept-2-yl)-but-3-en-2-one	Sesquiterpene
19	14.596	1.29	3,4,5,6,7,8-Hexahydro-1(2H)-naphthalenone	Quinone
20	14.650	0.40	3H-3,10A-methano-1,2-benzodioxocin-3-ol	Benzodioxocinol
21	15.118	4.65	1,3,4,5-tetrahydroxy-cyclohexanecarboxylic acid	Cyclohexanecarboxylic acid
22	15.890	0.24	Ingol-12-acetate	Terpenoids
23	16.305	1.19	(2E)-3,7,11,15-Tetramethyl-2-hexadecene	Hydrocarbon
24	16.548	2.06	2,2-Dimethyl-1-oxaspiro [2.5]octan-4-one	Oxaspiro epoxide
25	16.711	11.86	Pluchidiol	Phenolic compound
26	16.994	3.35	Cyclo(L-prolyl-L-valine)	Amino acid
27	17.355	1.67	Diundecyl phthalate	Phthalic acid ester
28	17.828	2.14	3,3-Dimethylglutaric acid	Fatty acids
29	17.975	1.16	l-Leucine, N-cyclopropylcarbonyl-, dodecyl ester	Amino acid ester
30	18.148	1.59	3-Isobutylhexahydropyrrolo[1,2-a]pyrazin	Alkaloids
31	18.347	4.08	n-Hexadecanoic acid	Fatty acids
32	20.051	5.51	Xanthaumin	Terpenoids
33	20.857	0.63	3-Methylene-bicyclo[3.2.1]oct-6-en-8-ol	Terpenoids
34	21.782	0.97	(+/−)-Marmesin	Benzofuranacrylic acid
35	22.283	0.71	3-Benzylhexahydropyrrolo[1,2-a]pyrazine-1,4-dione	Alkaloids
36	29.063	1.09	4,6-Cholestadien-3.beta.-ol	Steroids
37	29.345	1.36	Lup-20(29)-en-3.beta.-ol, acetate	Triterpenoids

## Data Availability

The data presented in this study are available on request from the corresponding author.

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
