# Peer review of "Ex Vivo Antiplatelet and Thrombolytic Activity of Bioactive Fractions from the New-Fangled Stem Buds of Ficus religiosa L. with Simultaneous GC-MS Examination"

_molecules, 2023, doi:10.3390/molecules28093918_

Round 1

Reviewer 1 Report

An interesting study, but the presentation is a bit confusing. 

1. The introduction needs to be expanded. It is implied that herbal drugs may have less side-effects or greater well-being - caution is indicated here. many of the chemo drugs used in cancer treatment were derived initially from plants... 

2. Was an untreated (normal saline?) control used for platelet aggregation. If so, please show the results in the table for us to see the baseline induced aggregation.  Comment on how the PRP was generated - rpm should be reported as g. How would you account for activation caused by the centrifugation caused by the generation of PRP?

3. Was the positive control of aspirin used at increasing concentrations as shown in the table? What is the maximal concentration of aspirin that ought to be used, or have been in other papers? 

4. How was clot lysis calculated/analysed? Any visuals should be included. 

5. Do the authors mean chain-disrupting 'embarrassment' of lipid peroxidation - bad translation? 

6. For the statistical analysis, the sample size is small (n=3) for some studies, and for others larger (n=6). Were normality tests done? 

7. Since the animals were given the extracts, why was the blood not drawn and platelets assessed directly after metabolism of the drug, to complement the in vitro findings? To what extent were the extracts purified for administration to animals and what controls (animal) were used?

8. The discussion and conclusions need to be expanded. Platelet activity beyond aggregation could be considered. The references need to be relooked and other work in understanding platelet activation/function and fibrin formation also be considered. Bit of a stretch to look for 'harmless' drugs of any kind. I think it is irresponsible to use the term and better suited for a magazine. 

Author Response

Comments and Suggestions for Authors

An interesting study, but the presentation is a bit confusing. 

  1. The introduction needs to be expanded. It is implied that herbal drugs may have less side-effects or greater well-being - caution is indicated here. many of the chemo drugs used in cancer treatment were derived initially from plants... 

Ans. Now it is expanded

  1. Was an untreated (normal saline?) control used for platelet aggregation. If so, please show the results in the table for us to see the baseline induced aggregation.  Comment on how the PRP was generated - rpm should be reported as g. How would you account for activation caused by the centrifugation caused by the generation of PRP?

Ans. Yes untreated (normal saline?) control used for platelet aggregation. It was used as blank and set zero at λmax 425 nm. PRP is produced from the rats blood. It is a concentration of platelets that circulate through the blood and are critical for blood clotting. Platelets and the liquid plasma portion of the blood contain various factors vital for the cell multiplication, recruitment, and specialization required for healing. Centrifuge the blood using a 'soft' spin. Transfer the supernatant plasma containing platelets into another sterile tube (without anticoagulant). Centrifuge tube at a higher speed (a hard spin) to obtain a platelet concentrate. The lower 1/3rd is PRP and upper 2/3rd is platelet-poor plasma (PPP).

  1. Was the positive control of aspirin used at increasing concentrations as shown in the table? What is the maximal concentration of aspirin that ought to be used, or have been in other papers? 

Ans. Aspirin were dissolved in normal saline in increasing concentrations 2, 4, 6, 8, 10, 12, 14, 16, and 18 μg/mL. Maximum dose is 20 μg/mL.

  1. How was clot lysis calculated/analysed? Any visuals should be included. 

Ans. The photograph of clot lysis is presented in supplementary file.

  1. Do the authors mean chain-disrupting 'embarrassment' of lipid peroxidation - bad translation? 

Ans. Now this word 'embarrassment' has been removed from the text.

  1. For the statistical analysis, the sample size is small (n=3) for some studies, and for others larger (n=6). Were normality tests done? 

Ans. For animal study, sample size should be larger (n=6). In the present experiment animal dosing carried out for toxicity study and six animal was used in each group. To evaluate the antiplatelet, antithrombotic, thrombolytic activity, blood sample was collected form animals and treat as in vitro study. Here (n=3) is not a sample size. It is the repetitions of same study at three times.  In vitro study usually the work is repeated for three time to calculate mean± S.D.

  1. Since the animals were given the extracts, why was the blood not drawn and platelets assessed directly after metabolism of the drug, to complement the in vitro findings? To what extent were the extracts purified for administration to animals and what controls (animal) were used?

Ans. In the present experiment animal dosing carried out for toxicity study only. To evaluate the antiplatelet, antithrombotic, thrombolytic activity, blood sample was collected form animals without any of drug dosing. So this is termed as Ex vivo antiplatelet and thrombolytic activity.

  1. The discussion and conclusions need to be expanded. Platelet activity beyond aggregation could be considered. The references need to be relooked and other work in understanding platelet activation/function and fibrin formation also be considered. Bit of a stretch to look for 'harmless' drugs of any kind. I think it is irresponsible to use the term and better suited for a magazine.

Ans. All these things has been covered according to the suggestions and highlighted.

Reviewer 2 Report

I have read the work entitled Ex vivo antiplatelet and thrombolytic activity of bioactive fractions from the new-fangled stem buds of Ficus religiosa L. with simultaneous GC-MS examination. reports an interesting work to be considered in Molecules after major revision noted below.

The abstract can be improve avoiding the excess of abbreviation and in “ The antiplatelet activity of CFFR, MFFR and standard drug Aspirin at 50 μg/mL was 54.32, 86.61, and 87.57%,”  respectively is missing

The Introduction does not provide sufficient background.

The design of the Work it's not appropriate (Methodology, Results, and discussion and finally conclusions)

the methods must be adequately described P y T are necessary in point 4.4 for extraction and concentration.

The originality of the study as well as result is not emphasized enough. Authors should emphasize the importance of their results in comparison with other similar published results. The main drawback is insufficient discussion of the results. In my opinion the results presented in this manuscript are worth for publishing

Author Response

Comments and Suggestions for Authors

I have read the work entitled Ex vivo antiplatelet and thrombolytic activity of bioactive fractions from the new-fangled stem buds of Ficus religiosa L. with simultaneous GC-MS examination. reports an interesting work to be considered in Molecules after major revision noted below.

The abstract can be improve avoiding the excess of abbreviation and in “ The antiplatelet activity of CFFR, MFFR and standard drug Aspirin at 50 μg/mL was 54.32, 86.61, and 87.57%,”  respectively is missing.

Ans. Now It is corrected and highlighted.

The Introduction does not provide sufficient background.

Ans. Now it is expanded

The design of the Work it's not appropriate (Methodology, Results, and discussion and finally conclusions)

Ans. The work is designed by study of several research paper. Now the procedure of antiplatelet activity is revised.

The methods must be adequately described P y T are necessary in point 4.4 for extraction and concentration.

Ans. Now It is modified.

The originality of the study as well as result is not emphasized enough. Authors should emphasize the importance of their results in comparison with other similar published results. The main drawback is insufficient discussion of the results. In my opinion the results presented in this manuscript are worth for publishing

Ans. Now It is revised and highlighted.

Reviewer 3 Report

I went through your manuscript entitled " Ex vivo antiplatelet and thrombolytic activity of bioactive fractions from the new-fangled stem buds of Ficus religiosa L. with simultaneous GC-MS examination". Through there are some important findings are there, overall the manuscript is needed a minor revisions. I have given my suggestions below.

1.       Line 24, explain the meanings of CFFR & MFFR.

2.       The Global Burden of Disease Study 2010, the data in 2010 is too old.

3.       Table 1, what does superscript “ns” mean?

4.       Line 112, why did Ascorbic acid and MFFR show three values on the inhibition of free radical scavenging activity? The values are not consistent with Figure 2, why?

5.       Subscript should be used for IC50.

6.        The header fonts in tables/figures should be uniform.

7.       The abbreviations should be noted for the first time, Hb, RBC, WBC, etc.

8.       It is difficult to determine the toxicity studies just by the RBC, organ weight, etc.

Author Response

Comments and Suggestions for Authors

I went through your manuscript entitled " Ex vivo antiplatelet and thrombolytic activity of bioactive fractions from the new-fangled stem buds of Ficus religiosa L. with simultaneous GC-MS examination". Through there are some important findings are there, overall the manuscript is needed a minor revisions. I have given my suggestions below.

  1. Line 24, explain the meanings of CFFR & MFFR.

Ans. Chloroform and methanol fractions of Ficus religiosa (CFFR & MFFR)

  1. The Global Burden of Disease Study2010, the data in 2010 is too old.

Ans. The recent report has been updated in the manuscript.

  1. Table 1, what does superscript “ns” mean?

Ans: “ns” denotes none significant value

  1. Line 112, why did Ascorbic acid and MFFR show three values on the inhibition of free radical scavenging activity? The values are not consistent with Figure 2, why?

Ans. Now corrected and highlighted

  1. Subscript should be usedfor IC50.

Ans. Now corrected and highlighted

  1. The header fonts in tables/figures should be uniform.
  2. Now corrected

  1. The abbreviations should be noted for the first time, Hb, RBC, WBC, etc.

Ans. Now corrected and highlighted

  1. It is difficult to determine the toxicity studies just by the RBC, organ weight, etc.

Ans. Yes, Further toxicity study in detailed we will  be perform in future.

Round 2

Reviewer 2 Report

The authors answered all the questions and changed the manuscript accordingly. Therefore, I think that the manuscript can be accepted for publication.

Author Response

Editor Comments to Author:

The chromatogram presented in figure 3 is too poor. The authors should include an analytically better chromatogram or transfer this figure to supplemental material.

Ans. The peaks in the chromatogram presented in Figure 3 are very close, so the appearance of the chromatogram is poor. All the phytocomponents that appeared in the chromatogram are tabulated in Table 5 and Fig.3. So this figure has been transferred to the supplemental material section.

In addition, the authors must describe exactly how the metabolites were identified - only by comparison with the NIST Spectrum Library.? With pure standards?

Ans.

Each compound's proof of identity was supported by a qualitative chemical test, comparing the retention time and spectra with reliable samples (Sigma-Aldrich) and software adopted to handle mass spectra in the NIST Mass Spectral Library Ver. 2.0 d [22].
